# MAP1LC3C repression reduces CIITA- and HLA class II expression in non-small cell lung cancer

Lydie M. O. Barbeau[1], Nicky A. Beelen[2,3], Kim G. Savelkouls[1], Tom G. H. Keulers[1], Lotte Wieten[3], Kasper M. A. Rouschop[1‡]*

1 Department of Radiation Oncology (Maastro), GROW - School for Oncology and Reproduction, Maastricht University Medical Center+, Maastricht, The Netherlands, 2 Department of Internal Medicine, GROW - School for Oncology and Reproduction, Maastricht University Medical Center+, Maastricht, The Netherlands, 3 Department of Transplantation Immunology, GROW - School for Oncology and Reproduction, Maastricht University Medical Center+, Maastricht, The Netherlands

‡ Lead contact.
* kasper.rouschop@maastrichtuniversity.nl

## Abstract

In the last decade, advancements in understanding the genetic landscape of lung squamous cell carcinoma (LUSC) have significantly impacted therapy development. Immune checkpoint inhibitors (ICI) have shown great promise, improving overall and progression-free survival in approximately 25% of the patients. However, challenges remain, such as the lack of predictive biomarkers, difficulties in patient stratification, and identifying mechanisms that cancers use to become immune-resistant ("immune-cold"). Analysis of TCGA datasets reveals reduced *MAP1LC3C* expression in cancer. Further analysis indicates that low *MAP1LC3C* is associated with reduced CIITA and *HLA* expression and with decreased immune cell infiltration. In tumor cells, silencing MAP1LC3C inhibits CIITA expression and suppresses HLA class II production. These findings suggest that cancer cells are selected for low MAP1LC3C expression to evade efficient immune responses.

## Authors summary

The introduction of immune checkpoint inhibitors has resulted in improved treatment of 15%-20% of lung cancer patients. However, biomarkers to identify immunotherapy responsive patients are an urgent unmet need. Reduced expression of HLA molecules aids tumor cells to evade the immune system and may be refractory to immunotherapy. We observed that low expression of the autophagy-related protein MAP1LC3C in cancers is associated with decreased CIITA and HLA class II expression and with reduced immune cell infiltration. MAP1LC3C may therefore be used as a biomarker to distinguish between immune responsive versus non-immune responsive tumors.

**Data availability statement:** All relevant data are within the manuscript and its Supporting Information files.

**Funding:** These studies received financial support from the Dutch Cancer Society (KWF grant 12,276 and UM2015-7735 to K.R.).

## Introduction

Over the past decade, deep and extensive comprehension of the genetic landscape in lung squamous cell carcinoma (LUSC) [1] has led to significant progress in therapy [2]. The introduction of immune checkpoint inhibitors (ICI) is the latest significant improvement and enhances LUSC patient overall survival (OS) and progression-free survival (PFS) [3]. However, immunotherapy effectiveness is restricted to 15 to 20% of patients and is associated with development of severe adverse effects in many patients [4,5]. Despite the enormous potential of immunotherapy, the key challenge remains to understand the mechanisms that determine immunotherapy failure and development of biomarkers that are predictive of response to ICI [6].

Effective immune recognition is correlated to neoantigen presentation on human leucocyte antigen (HLA) molecules. Therefore, to escape immune surveillance, tumor cells frequently decrease HLA surface expression.

HLA class I (HLA-I) molecules are expressed on all cells of the body, except on red blood cells. HLA class I primarily present peptides derived from intracellularly produced proteins. The HLA class I peptide complexes act as ligands for T cell receptors and killer-cell immunoglobulin like receptors (KIR), thereby influencing CD8 T cell and natural killer (NK) cell activation [7].

HLA class II (HLA-II) is constitutively expressed by professional antigen presenting cells (APCs), pulmonary and intestinal epithelial cells [8] and cancer cells [9]. In murine sarcoma, human breast cancer and melanoma, HLA class II expression is associated with increased tumor rejection, improved response to ICI in patients and improved prognosis after therapy [10–12]. Reduced HLA class II expression is observed in a variety of human tumors and contributes to dampening of CD4$^+$ T cell-dependent immune responses [9]. Expression of HLA class II is predominantly controlled through transcription via class II major histocompatibility transactivator (CIITA) [13,14] and is enhanced by interferon- γ (IFNγ) signaling [9,15,16]. In tumor cells, defects are mainly the result of structural abnormalities or epigenetic, transcriptional and post-translational regulation of HLA class II molecules [17].

Autophagy is a cellular catabolic recycling process to maintain cellular homeostasis [18–20]. In addition, autophagy dependent degradation of exogenous and endogenous antigens is required for CD4$^+$ helper and CD8$^+$ cytotoxic T cell stimulation [21–23] through cross-presentation of antigens, including tumor antigens [24,25], on HLA class I [26–28]. Autophagy related proteins control additional functions such as protein trafficking, secretion and degradation [29] dependent on members of the microtubule-associated protein 1 light chain 3/ γ-aminobutyric-acid-receptor-associated protein (MAP1LC3/GABARAP) family proteins. In mammalian cells, at least six members exist: MAP1LC3A, MAP1LC3B, MAP1LC3C, GABARAP, GABARAPL1, and GABARAPL2.

The best studied protein of the LC3/GABARAP family is MAP1LC3B. MAP1LC3B is involved in cargo selection/targeting/degradation and fusion of extending phagophore membranes during autophagy. Unlike MAP1LC3B, the exact role of MAP1LC3C is poorly understood. MAP1LC3C functions independently of canonical MAP1LC3B-mediated autophagy [30], is involved in mitophagy [31], TAX1BP1/NDP52-mediated xenophagy [32–36], viral dissemination [37], maintenance of functional endoplasmic reticulum (ER) exit sites and efficient ER export [38,39]. In tumor cells, MAP1LC3C binds to the ligand-activated oncogenic receptor c-Met for autophagic degradation, thereby mediating a tumor suppressive function [40–42]. In patients treated with chemo(radiation), low MAP1LC3C expression correlates with a better prognosis and overall survival (OS), especially in low grade glioma, lung adenocarcinoma and LUSC [43–45]. Counterintuitively, we and others observed that MAP1LC3C expression correlates with immune inhibitors, immune stimulators and HLA molecules, suggesting that the contribution of immunity does not add to therapy effectiveness for chemo(radiation) [45]. In contrast, MAP1LC3C may be used as a valuable biomarker to identify immune responsive

tumors. In this study, we show that low expression of MAP1LC3C is associated with low HLA class II expression and decreased infiltration by immune cells. *In vitro*, MAP1LC3C directly controls HLA class II expression through regulation of CIITA expression. Together, our data suggest that reduced MAP1LC3C expression by cancer cells results in immune evasion advantages and may be used as a biomarker to identify immune responsive NSCLC tumors.

## Materials and methods

### Patient cohorts

Gene expression profiles of normal and cancer tissue were extracted from LUSC, lung adenocarcinoma (LUAD), head and neck squamous cell carcinoma (HNSCC), esophageal carcinoma (ESCA), stomach adenocarcinoma (STAD), bladder carcinoma (BLCA), colon adenocarcinoma (COAD) and rectum adenocarcinoma (READ) cohorts from *The Cancer Genomic Atlas* (TCGA) using the portal of the Santa Cruz University of California (https://xenabrowser.net/) [46]. Gene expression is measured as FKPM, Fragments per Kilobase of transcript per million Mapped reads. Gene signatures for myeloid and lymphoid immune cell subgroups were generated by extracting differentially expressed genes (DEGs) list from a leucocyte gene signature matrix (LM22) developed by Newman et al. (GSE65136) [47] (Table S3). Patient cohorts were filtered for availability of *MAP1LC3C* expression data and expression data of primary tumor vs solid normal tissue. Human papilloma virus (HPV) positive patients were excluded from the HNSCC cohort. Single-cell transcriptomic sequencing (scRNA seq) data from a total of 40,362 cell transcriptomes from seven patients undergoing resection of non-small lung cancer NSCLC were extracted from Zilionis et al. (GSE127465) [48]. Gene expression is measured as TPM, Transcripts Per Million.

### Ethics statement

The current study only uses publically available, open-access, datasets provided by the *The Cancer Genomic Atlas* (TCGA) [46] and Zilionis et al. (GSE127465) [48]. As such, ethical approval was already obtained for the original publications and no additional ethical approval was applied for.

### Cell line and cell culture

Male lung squamous cell carcinoma, SW900, were cultured in RPMI-1640 (Sigma Aldrich) supplemented with 10% FBS. Cells were regularly tested for mycoplasma contamination.

### Stable cell line generation

To achieve constitutive deficiency of MAP1LC3C, lentiviral-mediated knockdown was performed using short hairpin RNA (shRNA) vectors. Specifically, shRNA constructs targeting MAP1LC3C were expressed from pLKO.1 vectors (Sigma-Aldrich, TRCN0000163943 and TRCN0000159061) (Table 1). For doxycycline-inducible CIITA expression, myc-tagged CIITA was transferred from pcDNA3 myc-CIITA (addgene, #14650[49]) to pCW57-MCS1-P2A-MCS2-myc-CIITA using EcoRI restriction. Correct orientation and DNA sequence was verified through Sanger-sequencing.

**Table 1. MAP1LC3C shRNA sequences.**

| TRC number | shRNA sequences | Targeting location |
|---|---|---|
| TRCN0000163943 (sh#1) | GTGTTCTTGTGTCCTTCTA | Exon 4 |
| TRCN0000159061 (sh#2) | TGCTCTTGAAAGTTATATA | Exon 4 |

Lentiviral particles were produced by co-transfecting HEK293T cells with the respective transfer vector, along with the packaging plasmids psPAX2 (Addgene #12260) and pCMV-VSV-G (Addgene #8454) using PPEi transfection reagent. After 48 hours, viral supernatants were collected and filtered through a 0.45 μm syringe filter. Target cells were transduced by incubating them with viral supernatants in the presence of 20 μg/mL Protamine Sulfate to enhance viral entry.

Following 48 hours of transduction, stable knockdown cells were selected using 1 μg/mL puromycin for at least 7 days. For CIITA overexpression, cells were cultured for 48 hours post-transduction before treatment with 1 μg/mL doxycycline to induce CIITA expression.

## Drugs and compounds

Chloroquine (CQ) (Sigma-Aldrich, cat#C6628) and Doxycycline (Sigma-Aldrich, cat# D9891) were dissolved in water. Tazemetostat (MedChemExpress, cat# HY-13803), Vorinostat (Cayman Chemical, cat# 10009929 and Romidepsin (Cayman Chemical, cat# 17130) were dissolved in DMSO. Vorinostat and Romidepsin used were kind gifts from K.R. Kampen (KU Leuven, Belgium). Cells were once treated with Interferon gamma-1b (IFNγ, 50ng/mL) (IMMUKIN, EMC, cat# 774448), CQ (5μg/mL), Tazemetostat (10 μM), Vorinostat (1 μM) or Doxycycline (1μg/mL) one day after seeding. Unless indicated otherwise and when indicated, cells were exposed for 24 hours to IFNγ, CQ, Tazemetostat or Vorinostat. Doxycycline exposure was done for 2 days before experimentation to achieve sufficient knockdown..

## Immunoblotting

Cells were rinsed twice with PBS and directly lysed in RIPA buffer (50mM Tris-HCl, 150mM NaCl, 0.1% Triton X100, 0.5% Sodium deoxycholate, 0.1% SDS, 1mM Sodium orthovanadate, 1mM NaF, protease inhibitors (Roche pill Complete Inhibitor), pH 8.0). Protein concentrations were determined with Bradford Protein Assay (Bio-Rad, cat# 5000006). Proteins were separated on Tris-HCL SDS-PAGE gels and transferred onto PVDF membranes. Membranes were blocked for 60 minutes in 5% casein (Sigma-Aldrich, cat# C3400) and 0.1% Tween20 in 1X PBS. Membranes were probed overnight at 4°C with primary antibodies and visualized using appropriate HRP-linked secondary antibodies (Table 2). Amersham ECL primer Western Blotting Detection Reagent (GE Healthcare) was used for visualization and imaging was performed with the Azure 600 (Azure Biosystems). Immunoblots were quantified using Image J [50].

## Quantitative PCR

Total RNA was isolated from cells using NucleoSpin RNA kit (Macherey Nagel, cat# MN 740955.250) according to the manufacturer's protocol. Relative abundance was determined

**Table 2. Primary and secondary antibodies for immunoblotting.**

| Antibody | Conjugate | Species | Conc./Dil. | Company | Cat# |
|---|---|---|---|---|---|
| Primary | | | | | |
| Anti-LC3B | – | Rabbit | 51ng/mL | Cell Signaling | 2775S |
| Anti-DP/DQ/DR CR3 /43 | – | Mouse | 77ng/mL | Dako | M077501-2 |
| Anti-actin clone C4 | – | Mouse | – | MP Biomedicals | 691001 |
| Secondary | | | | | |
| Anti-mouse | HRP | Horse | 30.6ng/mL | Cell Signaling | 7076S |
| Anti-rabbit | HRP | Goat | 13.14ng/mL | Cell Signaling | 7074S |

after cDNA synthesis (Biorad, Iscript) followed by target specific (Table 3) SYBR-green (Sensi-Mix SYBR high-ROX, GC Biotech) based quantitative PCR (qPCR).

*MAP1LC3C* and *18S* mRNA expression were analysed using TaqMan gene FAM/MGB-NFQ probes (Table 4). MAP1LC3C cDNA was pre-amplified using TaqMan PreAmp Master mix (Thermo Fisher, cat# 4391128) followed by TaqMan probe-based RT-qPCR (Thermo Fisher, cat# 4444556). *RPL13A* and *18S* were used as housekeeping genes.

## Immunofluorescence microscopy

For HLA class II staining, cells were fixed in 4% paraformaldehyde (PFA) for 15 minutes on ice. After permeabilization (0.5% Tween20 in PBS) for 20 minutes at room temperature (RT) and blocking (5% normal goat serum, 0.05% Tween20 in PBS) for 30 minutes at RT, cells were incubated with mouse anti-HLA/DP/DQ/DR in blocking buffer for 90 minutes at RT. Antibody binding was visualized with goat anti-mouse Alexa488. Nuclei were identified with Hoechst 33342 (Sigma-Aldrich, cat# B2261) and analyzed using a Leica TCS SP8 confocal microscope and LAS X software.

## HLA cell surface expression

Tumor cells were treated with 50ng/mL IFN-γ, 24h prior harvest. Cell surface staining were performed in PBS using a combination of anti-HLA antibodies and Live/Dead˙ Fixable Aqua

**Table 3. Quantitative real-time PCR primer sequences.**

| Target genes | Forward (5' – 3') | Reverse (5' – 3') |
|---|---|---|
| *RPL13A* | CCGGGTTGGCTGGAAAGGTAATTATG | CTTCTCGGCCTGTTTCCGTAC |
| *HLA-DPA1* | GAGCTGTGATCTTGAGAGC | CTGTTGGTCTATGCGTCTGTAC |
| *HLA-DMA* | GTGTGGCAAGAAGGTATGGG | GTCATCTGGCCACATTGGAGT |
| *HLA-DRA* | GCCATAAGTGGAGTCCCTGT | CGCCTGATTGGTCAGGATTC |
| *HLA-DPB1* | GCTCTGACGGCGTTACTGAT | GCGCTGTGTCCCATTAAACG |
| *HLA-A* | ACAGACTGACCGAGTGGA | CACGTCGCAGCCATACATTATC |
| *HLA-B* | TAGCAGTTGTGGTCATCGGA | ACAGCTGTCTCAGGCTTTTCAA |
| *HLA-C* | ACACAGAAGTACAAGCGCCA | CGTAGGCGGACTGGTCATAC |
| *CIITA* | CCTGGAGCTTCTTAACAGCGA | TGTGTCGGGTTCTGAGTAGAG |
| *PD-L1* | GGCATTTGCTGAACGCATTT | GGTCTTCCTCTCCATGCACAA |
| *PD-L2* | ACCAGTGTTCTGCGCCTAAA | CCTGGGTTCCATCTGACTTTGA |
| *CXCL10* | TGATGGCCTTCGATTCTGGATT | GTGGCATTCAAGGAGTACCTC |
| *IL1A* | AGATGCCTGAGATACCCAAAACC | CCAAGCACACCCAGTAGTCT |
| *IL6* | CCTGAACCTTCCAAAGATGGC | TTCACCAGGCAAGTCTCCTCA |
| *STAT1* | GGCAAAGAGTGATCAGAAACAA | GTTCAGTGACATTCAGCAACTC |
| *MAP1LC3B* | AACGGGCTGTGTGAGAAAAC | AGTGAGGACTTTGGGTGTGG |
| *MAP1LC3C* | AAACCAAGTTCCTGGTCCCG | ACACGAAGCCATCCTCATCC |
| *GATA4* | GTGTCCCAGACGTTCTCAGTC | GGGAGACGCATAGCCTTGT |
| *c-Myc* | GGTCTTTTCATTGTTTTCCA | TCAAGAGGCGAACACACAAC |

**Table 4. TaqMan probes.**

| Target genes | Assay ID TaqMan |
|---|---|
| *18S* | Hs03928990_g1 |
| *MAP1LC3C* | Hs01374916_m1 |

Dead Cell Stain Kit. For total HLA-expression, cells were fixed with 1% PFA. Analysis was performed on the viable cell population. All FACS data were acquired on a BD FACS Canto II (BD Biosciences) and analyzed with FlowJo v10.8.1 software.

### Intracellular and cell surface IFNγ receptor (IFNgR) assay

Tumor cells were treated with 50ng/mL IFNγ 24h prior harvest. Cells were stained with LIVE/DEAD Aqua dye and subsequently equally divided. For total levels, cells were fixed and permeabilized with fixation/permeabilization kit. After permeabilization, cells were probed with anti-CD119/IFNgR1 (Table 5) in permeabilization buffer on ice. Cell surface expression was determined without fixation and permeabilization in viable cells (LIVE/DEAD Aqua dye). Cells were temporarily stored in 1% PFA/PBS solution until flow cytometry analysis was performed.

### Statistics

Data are expressed as mean ± SD, unless indicated differently. Statistical analyses were performed using GraphPad prism software. Mann-Whitney t-test and unpaired t-test were used for comparison between two groups. Spearman correlation (r) values were calculated for association between *MAP1LC3C* expression and genes' expression for the TCGA data LUSC, LUAD and HNSCC cohorts. Pearson correlation (r) values were calculated for association between *MAP1LC3C* expression and other genes' expression for lung tumors patients. Ordinary one-way ANOVA was used for comparisons between control group against all other groups. Statistical significant differences are indicated as *p < 0.05, **p < 0.01, ***p < 0.001 or ****p < 0.0001.

## Results

### Decreased MAP1LC3C expression correlates with low immune cell infiltration and reduced HLA class II expression

Analysis of the gene expression of the lung squamous cell carcinoma (LUSC) cohort from The Cancer Genome Atlas (TCGA) revealed a significant decrease in *MAP1LC3C* mRNA expression in tumors (Fig 1A). In LUSC, *MAP1LC3C* expression ranged from levels observed in normal tissue to undetectable levels in 5.2% of the patients. Further analysis revealed a consistent overall reduction in *MAP1LC3C* expression across a minimum of seven cancer types, including esophageal carcinoma, stomach adenocarcinoma, bladder carcinoma, colon adenocarcinoma, rectum adenocarcinoma, lung adenocarcinoma and head and neck squamous cell carcinoma (Fig. S1A-G), suggesting a selection advantage for tumor cells with decreased *MAP1LC3C* expression.

**Table 5. Primary and secondary antibodies for flow cytometry analysis.**

| Antibodies and dyes | Conjugate | Isotype | Conc./Dil. | Company | Cat# |
|---|---|---|---|---|---|
| Primary | | | | | |
| HLA-DR | PerCP-Vio700 | REA805 | 5µg/mL | Miltenyi | 130-11-793 |
| HLA-DQ/DP/DR CR3 /43 | Purified | IgG | 3.08µg/mL | Dako | M0775 |
| CD119/IFNgR1 | APC | REA161 | 30 µg/mL | Miltenyi | 130-125-846 |
| Dyes | | | | | |
| Live/Dead˙ Fixable Dead Cell Stain Kit | Aqua | – | – | Thermo Fisher | L34957 |
| Secondary | | | | | |
| Goat anti-mouse Ig | FITC | IgG | 2µg/mL | BD | 554001 |
| Isotypes | | | | | |
| REA | PerCP-Vio700 | REA | 2µg/mL | Miltenyi | 130-104-620 |

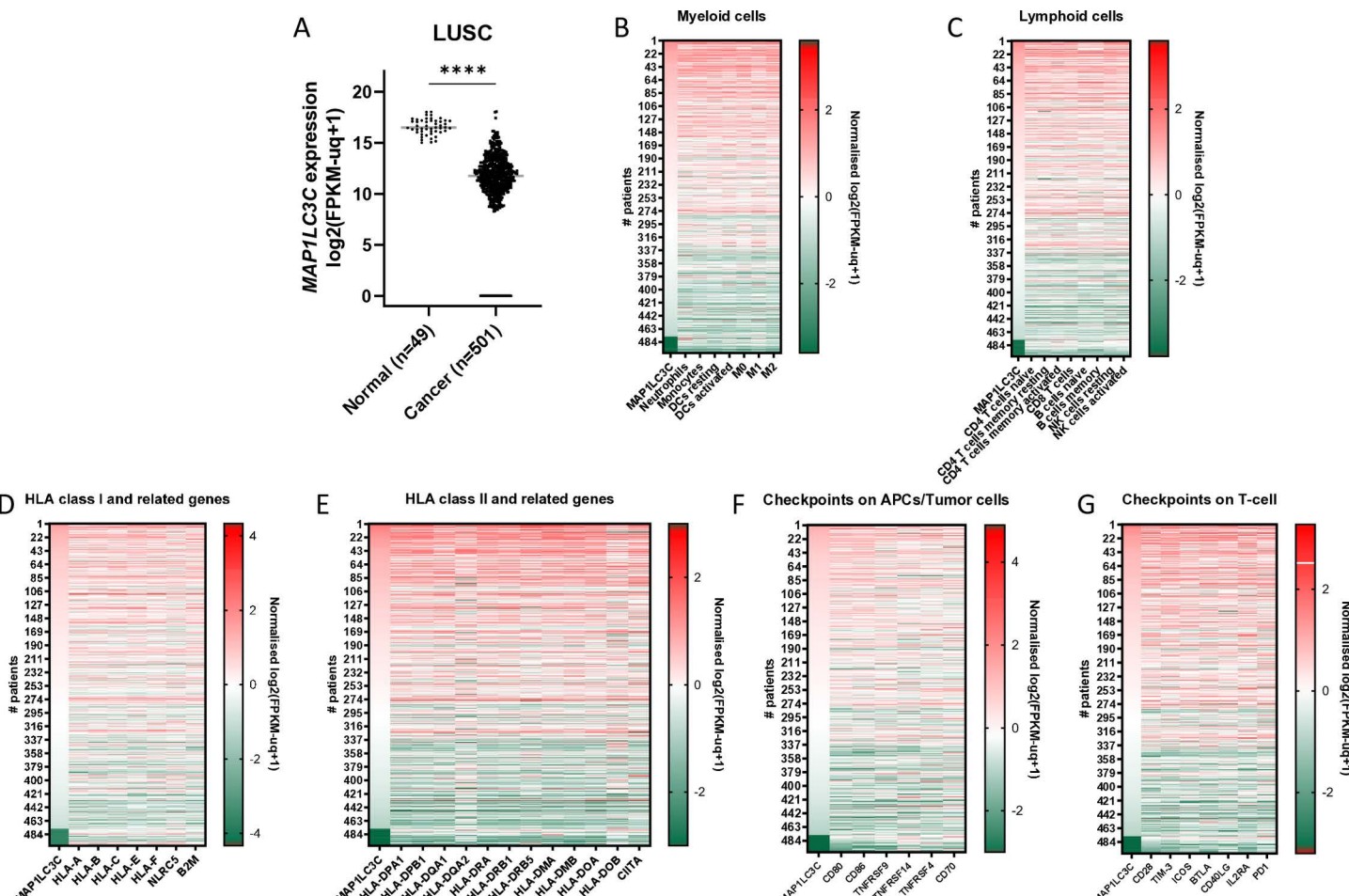

**Fig 1. Low *MAP1LC3C* expression in LUSC correlates with low cancer immunity.** A) *MAP1LC3C* expression is decreased in tumors (n = 501) compared to normal lung tissue (n = 49) (median in grey, ****p < 0.0001). Heat maps ranked for *MAP1LC3C* expression for B) myeloid and C) lymphoid immune cells' gene signatures, D) HLA class I and E) HLA class II, HLA chaperones and *CIITA*, F) immune checkpoints on APCs/tumor cells and G) immune checkpoints on T-cells. FKPM, Fragments per Kilobase of transcript per million Mapped reads. Spearman (r) correlation are indicated in Table 6.

A general phenomenon for cancer development is the selection of tumor cells with the ability to escape immune surveillance and results in dampened immune activation and/or immune infiltration. To determine if an association exists between immune-cell infiltration and *MAP1LC3C* expression, expression patterns with immune cell identifiers were assessed. Using established leucocyte gene signatures [47] (Table S3), our analysis revealed a correlation between *MAP1LC3C* expression and immune cell infiltration, i.e., myeloid cells (neutrophils, Spearman r = 0.615; monocytes, r = 0.6732; DCs resting, r = 0.6985; DCs activated, r = 0.6351; macrophages, 0.5774 < r < 0.7107) and lymphoid cells (CD4 T cells; 0.5143 < r < 0.5834; CD8 T cells, r = 0.5453; B cells naïve, r = 0.5722; B cells memory, r = 0.5674; NK cells resting, r = 0.5861; NK cells activated, r = 0.5241) (Fig 1B, C and Table 6).

These results indicate an association between MAP1LC3C and immune responsiveness and/or recognition.

Reduced antigen presentation and increased immune checkpoint activation contribute to immune recognition avoidance. Therefore, we assessed the correlation between MAP1LC3C mRNA expression and the gene expression of the known and available HLA.

**Table 6. Spearman correlation (r) values between *MAP1LC3C* and cancer immunity genes expression are indicated in LUSC. \* : 0.5 < r < 0.6; \*\* : r > 0.6., all p-values are < 0.0001.**

| Myeloid progenitor cells | Neutrophils | Monocytes | DCs resting | DCs activated | M0 | M1 | M2 | |
|---|---|---|---|---|---|---|---|---|
| Spearman r | 0.6150** | 0.6732** | 0.6985** | 0.6351** | 0.6902** | 0,5774* | 0.7107** | |
| Lymphoid progenitor cells | CD4 T cells naïve | CD4 T cells memory resting | CD4 T cells memory activated | CD8 T cells | B cells naïve | B cells memory | NK cells resting | NK cells activated |
| Spearman r | 0.5811* | 0.5834* | 0.5143* | 0.5453* | 0.5722* | 0.5674* | 0.5861* | 0.5241* |
| HLA class I and related genes | HLA-A | HLA-B | HLA-C | HLA-E | HLA-F | NLRC5 | B2M | |
| Spearman r | 0.4153 | 0.4762 | 0.4708 | 0.5075* | 0.4198 | 0.2918 | 0.4902 | |
| HLA class II and related genes | HLA-DPA1 | HLA-DPB1 | HLA-DQA1 | HLA-DQA2 | HLA-DRA | HLA-DRB1 | HLA-DRB5 | |
| Spearman r | 0.6494** | 0.6592** | 0.6009** | 0.4597 | 0.661** | 0.6355** | 0.574* | |
| | HLA-DMA | HLA-DMB | HLA-DOA | HLA-DOB | CIITA | | | |
| | 0.6337** | 0.6565** | 0.6108** | 0.3791 | 0.5101* | | | |
| Checkpoints on APCs and tumor cells | CD80 | CD86 | TNFRSF9 | TNFRSF14 | TNFRSF4 | CD70 | | |
| Spearman r | 0.608** | 0.6141** | 0.4747 | 0.3437 | 0.4283 | 0.3032 | | |
| Checkpoints on T-cell | CD28 | TIM-3 | ICOS | BTLA | CD40LG | IL2RA | PD1 | |
| Spearman r | 0.6257** | 0.6387** | 0.5498* | 0.515* | 0.561* | 0.5780* | 0.4566 | |

We observed a moderate association between *HLA class I* and *MAP1LC3C* expression (HLA-A, r = 0.4153; HLA-B; r = 0.4762; HLA-C, r = 0.4708; HLA-E, r = 0.5075; HLA-F, r = 0.4198) (Fig 1D, Table 6). Remarkably, a stronger association between, *MAP1LC3C* expression, *HLA class II* genes (HLA-DPA1, r = 0.6494; HLA-DPB1, r = 0.6592; HLA-DQA1, r = 0.6009; HLA-DQA2, r = 0.4597; HLA-DRA, r = 0.661; HLA-DRB1, r = 0.6355; HLA-DRB5, r = 0.574), HLA chaperone genes (HLA-DMA, r = 0.6337; HLA-DMB, r = 0.6565; HLA-DOA, r = 0.6108)and the major histocompatibility complex (MHC) class II transactivator (CIITA, r = 0.5101) were observed (Fig 1E). In addition, MAP1LC3C correlates with tumor-cell associated CD80 and CD86, cell costimulatory surface proteins that are primarily expressed on antigen-presenting cells (APCs) and tumor cells (Fig 1F). In line with reduced T-cell infiltration (Fig 1C), several immune checkpoints correlate with *MAP1LC3C* expression (Fig 1G, Table 6). Strikingly there is a consistent positive correlation between both co-stimulatory receptors and T-cell receptors and *MAP1LC3C* expression, regardless of whether they are inhibiting or activating signaling, suggesting a restrictive immune response environment. Although comparable associations are observed in LUAD (Fig S1H-M, Table S1) and HNSCC patients (Fig S1N-S, Table S2), correlations with HLA and checkpoint expression are less pronounced and suggest that, in these cancer types, additional mechanisms are at play. In summary, this data suggests that LUSC with reduced *MAP1LC3C* expression are "immune cold".

Consistent with these observations, single-cell transcriptomic analysis of a small cohort of 7 NSCLC tumors (GSE127465) [48] confirmed that *MAP1LC3C* expression is associated with *HLA class II*, HLA chaperone (HLA-DMA) and *CIITA* (Fig 2A) expression. Importantly, *MAP1LC3C* expression is detected in all cell populations, excluding red blood cells (Fig 2B). Notably, the highest average expression of *MAP1LC3C* is observed in B-cells (0.0047 TPM, transcripts per million) and cancer cells (0.0036 TPM) with a large variation in expression between tumors (Fig 2B), suggesting that inter-cancer *MAP1LC3C* differences, as observed in TCGA data, are driven by expression differences in cancer cells.

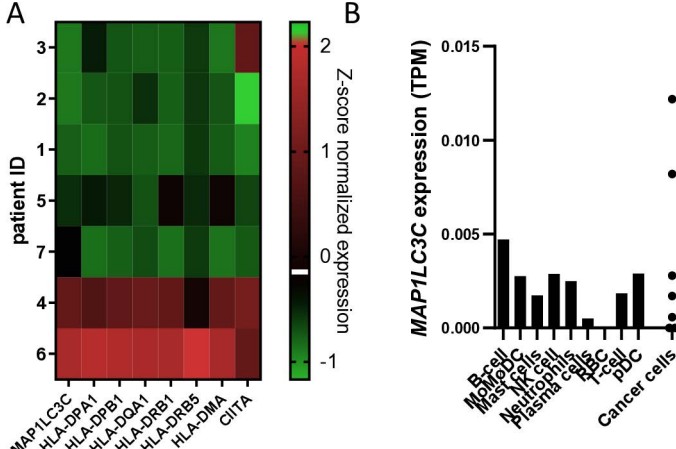

**Fig 2. Cancer cells *MAP1LC3C* expression correlates with *HLA class II* gene expression.** Heat map of *MAP1LC3C* expression and *HLA class II* (*HLA-DPA1, HLA-DPB1, HLA-DQA1, HLA-DRB1 and HLA-DRB5*), HLA chaperone (*HLA-DMA*) and *CIITA* (left to right) expression (Z-score normalized data). B) Average mRNA expression level obtained from single-cell transcriptomics of NSCLC human tumors analysis. Data were extracted from lung squamous cell carcinoma (n = 2) and lung adenocarcinoma (n = 5) (MoMØDC, Monocytes - Macrophages - Dendritic cells; RBC, red blood cells; pDC, plasmacytoid DCs). TPM, Transcripts Per Million.

## MAP1LC3C controls CIITA and HLA class II expression

To test whether MAP1LC3C expression directly controls *HLA class II* expression, MAP1LC3C deficient SW900 (LUSC) cells were generated and assessed for MAP1LC3C mRNA (Fig 3A) and protein expression (Fig 3B). Further mRNA analyses revealed that MAP1LC3C deficiency resulted in decreased expression of HLA class II mRNA (*HLA-DRA, HLA-DPA1, HLA-DPB1*) (Fig 3C-E), HLA chaperone gene, *HLA-DMA* (Fig 3F) but not *HLA class I* genes (data not shown*). Other HLA were not detectable in this cell line. The general effect on HLA class II expression suggests that MAP1LC3C influences a common upstream denominator. In line, the essential transcription factor for HLA class II expression, *CIITA,* is dependent on MAP1LC3C expression (Fig 3G).

Importantly, HLA class II expression on cancer cells increases due to IFNγ signaling [9,15]. As expected, HLA class II protein expression is increased after interferon-γ (IFNγ) stimulation in control cells as assessed by immunofluorescent staining and flow cytometry (Fig 3H, I and J). Yet, in comparable to abrogated basal expression in MAP1LC3C deficient cells, IFNγ stimulation fail to express HLA class II to comparable levels as control cells as determined by immunofluorescent microscopy (Fig 3H, J and K).

MAP1LC3C is a homologue of MAP1LC3B, an essential protein for execution of autophagy. To exclude that enhanced autophagic/lysosomal HLA class II degradation results in decreased expression in MAP1LC3C deficient cells, HLA class II degradation was determined. As suc,h degradation was inhibited by a sufficient dose chloroquine (CQ)) to block autophagy flux, as illustrated by LC3B-II accumulation (Fig S2J), or after genetic targeting autophagy (MAP1LC3B) [51]. These data indicate that the observed HLA class II expression differences are not the result of increased lysosomal degradation (Fig 3K and S2J), but due to decreased mRNA production (Fig 3C-G).

## MAP1LC3C controls HLA class II expression through CIITA regulation

These observations prompted us to evaluate if MAP1LC3C controls HLA class II expression through inhibition of IFNγ-signaling. We observed no differences in the total level or cell

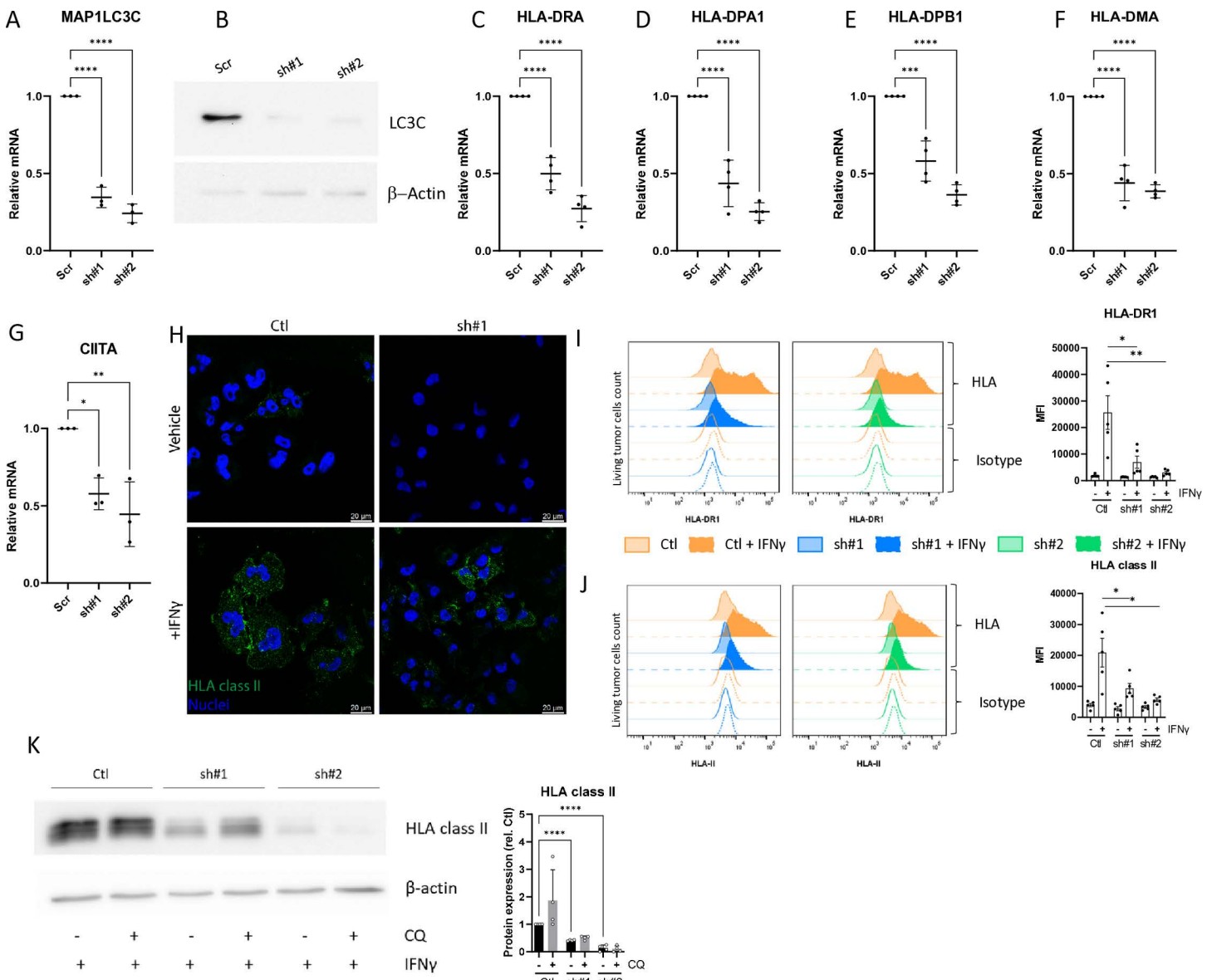

**Fig 3. Loss of MAP1LC3C impairs basal and IFN γ-induced HLA class II expression.** A) Quantitative PCR of *MAP1LC3C* in control and 2 independent SW900 MAP1LC3C knockdown cell lines. B) Confirmation of MAP1LC3C knockdown by immunoblot analysis C-G) Quantitative PCR of *HLA class* **II** *(HLA-DRA, HLA-DPA1, HLA-DPB1)*, *HLA chaperone* *(HLA-DMA)* and *CIITA* mRNA expression in control and 2 independent MAP1LC3C deficient SW900 cell lines (*p < 0.05, **p < 0.01, ***p < 0.001, ****p < 0.0001). H) Immunofluorescent staining of pan-HLA class **II** (HLA-DP/DQ/DR) in SW900 control cells or MAP1LC3C deficient cells following IFNγ treatment (50 ng/ml, 24h). Scale bar: 36.8μM. I) Histograms of flow cytometry analysis and quantification of HLA-DR1 and J) pan-HLA class II protein cell surface expression following IFNγ treatment (*p < 0.05, **p < 0.01). K) Immunoblot analysis and quantification of pan-HLA class **II** (HLA-DP/DQ/DR) after IFNγ (50 ng/ml, 24h) and CQ (5 μg/ml, 24h) exposure (****p < 0.0001). J-K) Data are representative of three or more independent experiments and values are expressed as mean ± SD (A-I) and as median ± SD (J-K).

surface expression of the IFNγ receptor (Fig S3A-B). In addition, IFNγ stimulation resulted in *STAT1* and JAK-STAT-mediated gene expression (PD-L1, PD-L2, IL1a, IL6 and CXCL10) independent on MAP1LC3C expression (Fig S3C-H). These results indicate that silencing MAP1LC3C does not impair IFNγ receptor activation or inhibition of downstream signaling.

We observed that MAP1LC3C deficiency results in impaired CIITA expression (Fig 3G). We therefore evaluated whether MAP1LC3C affects HLA class II expression through direct

inhibition of CIITA expression or through inhibition of CIITA-mediated effects (e.g., missing cofactor) [52,53]. For this purpose, cell lines with, doxycycline inducible, ectopic CIITA expression were engineered. Although there is a minimal leakiness in the absence of doxycycline, addition of doxycycline elevated CIITA expression in the absence of MAP1LC3C (Fig 4A and B). Importantly, ectopic CIITA expression rescued *HLA class II* and HLA chaperone (*HLA-DMA*) mRNA, and protein expression in MAP1LC3C deficient cells (Fig 4A-G). These data indicate, that although MAP1LC3C deficiency results in decreased HLA class II expression, these cells are intrinsically capable CIITA-induced gene regulation. Furthermore, these data suggests that MAP1LC3C regulates CIITA expression at the transcriptional level. Doxycycline alone had no effect on HLA class II, HLA chaperone or CIITA expression (Fig S4A-E).

We next aimed to explore alterations in MAP1LC3C deficient cells that control CIITA expression. CIITA expression is, at least in part, regulated by Polycomb Repressive Complex 2 (PRC2) [62]. PRC2, particularly through its catalytic subunit EZH2, represses CIITA transcription by methylating histone H3 at lysine 27 (H3K27me3) at the CIITA promoter, especially the IFN-γ-inducible promoter IV[54]. This repressive histone modification leads to chromatin compaction, limiting the transcriptional activation of CIITA and, subsequently, MHC-II gene expression. Hence, we evaluated whether MAP1LC3C may be required to alleviate PRC2-mediated CIITA silencing. Although exposure to the PRC2 inhibitor, Tazemetostat, induced PRC2 target gene expression in control cells, no rescue was observed of PRC2 target genes, *HLA class II*, HLA chaperone and *CIITA* in MAP1LC3C deficient cells indicating that MAP1LC3C does not control CIITA expression through PRC2 (Fig S4F-J).

In addition, deacetylation of histones by HDACs tightens histones interaction with DNA, resulting in gene transcription impairment [55]. HLA class II transcription is dependent on recruitment of histone modifier enzymes, such as histone deacetylases (HDACs)[56]. Analysis of the proximal interactome network [57,58] revealed that CHAF1A, TRIM28 and PCNA interact with MAP1LC3C. These proteins recruit HDACs [59,60] and modify HDAC activity [61], repetively.

Therefore, we determined whether HDAC suppress MAP1LC3C-dependent HLA class II gene expression [63,64]. Treatment of MAP1LC3C deficient cells with a histone deacetylase inhibitor, vorinostat, resulted in increased mRNA expression of the known HDAC inhibitor target gene, *c-Myc* (Fig 4H) [65] and restored HLA class II expression (Fig 4I-K), independent of *CIITA* expression (Fig 4L). HLA class II immunoblot analysis confirmed the elevated expression in MAP1LC3C deficient cells (Fig 4M). Together, these results indicate that, in the absence of MAP1LC3C, intrinsic mechanisms that are required for HLA transcriptional expression are intact but are repressed subsequently to MAP1LC3C-mediated CIITA loss.

## Discussion

Despite the development of immune checkpoint inhibitors, many tumors relapse or fail to respond to therapy. Therefore, a key challenge is to predict responders vs non-responders. Essential in responsiveness of cancers to immune therapy is the extent of immune cell infiltration, recognition and immune activation. Here, we show that low expression of MAP1LC3C, a non-canonical autophagy protein, defines a subgroup of tumors with reduced *HLA class II* expression and low immune cell infiltration. Evaluation *in vitro*, revealed that MAP1LC3C loss prevents *CIITA* expression, the key regulator of HLA class II transcription.

Tumor cells are continuously selected for properties that allow escape from immune surveillance, resulting in cancer phenotypes that are challenging to eliminate for immune cells [67]. Various mechanisms have been described that contribute to immunoediting, including selection of variants that display fewer immunogenic antigens [68], reduced tumor-specific

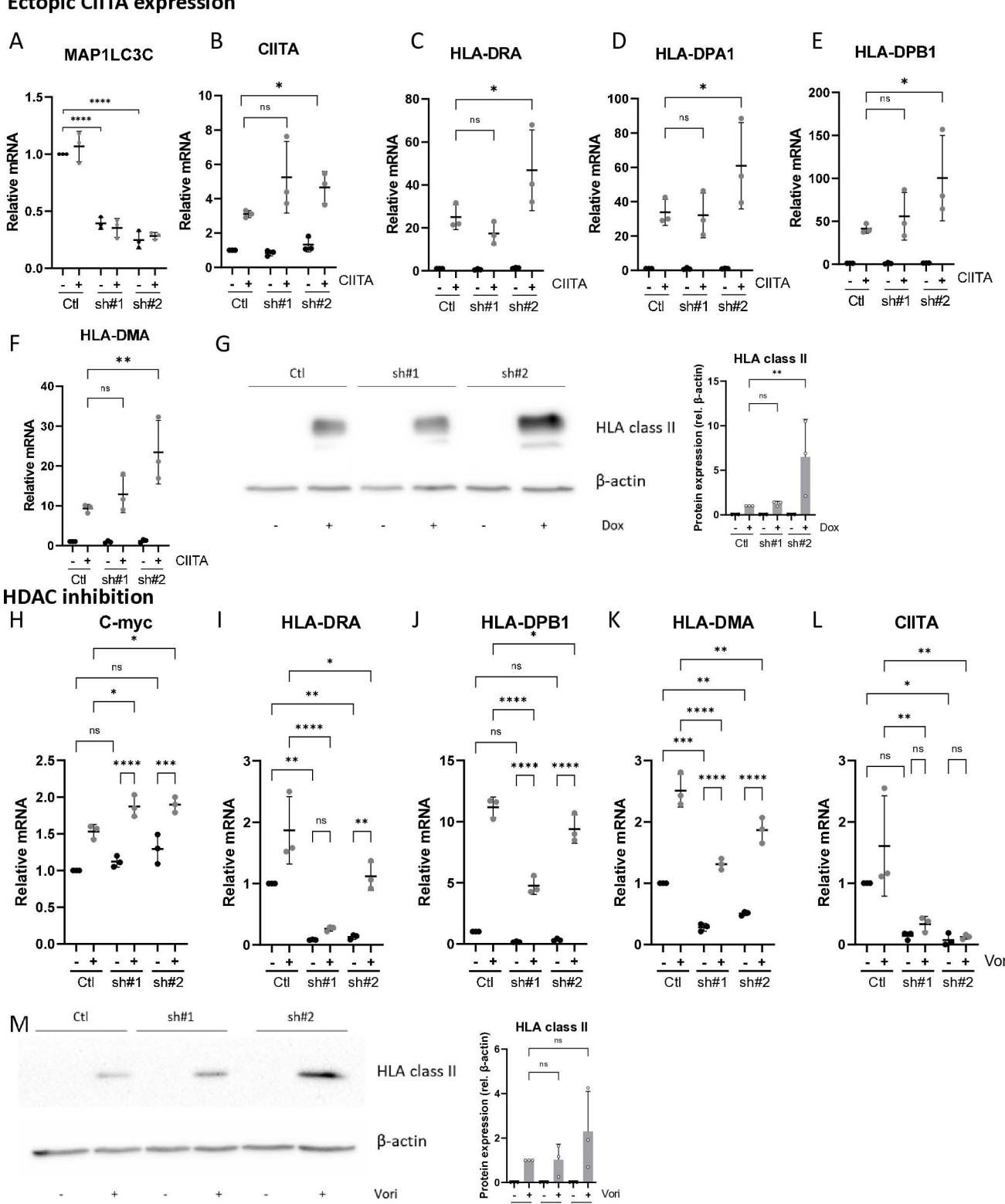

**Fig 4. Ectopic CIITA expression and HDAC inhibition rescues HLA class II expression in MAP1LC3C deficient SW900 cells.** Quantitative PCR of A) MAP1LC3C mRNA in engineered cells, with and without exposure to doxycycline/CITTA-induction (doxyxycline 1μg/mL, 48h). B) *CIITA* and C-E) *HLA class II* and F) HLA chaperone (*HLA-DMA*) expression following CIITA overexpression. (ns: non-significant, *p < 0.05); **p < 0.01).

G) Immunoblot analysis and quantification of pan-HLA class II protein levels following CIITA overexpression (**$p < 0.01$). Quantitative PCR of H) HDAC target gene *(c-Myc)*, I-J) *HLA class II*, K) HLA chaperone *(HLA-DMA)* and L) *CIITA* mRNA expression following HDAC inhibition (Vorinostat 1 µM, 24h) (ns: non-significant, *$p < 0.05$, **$p < 0.01$, ***$p < 0.001$, ****$p < 0.0001$). M) Immunoblot analysis and quantification of pan-HLA class **II** (HLA-DP/DQ/DR) protein following HDAC inhibition (Vori).Data are representative of three or more independent experiments and values are expressed as mean ± SD.

neoantigens [69], shielding via extracellular matrix deposition [70], acidification [71], expression of inhibitory immune checkpoints molecules or their receptors [72] and down-regulation of HLA [73]. Reduced HLA expression is thought to play a major role in the ability to evade immune responses[73]. HLA class I is ubiquitously expressed on most cell types, whereas HLA class II is typically restricted to specialized antigen presenting cells and tumor cells. HLA class II expression is observed in glioma, lung cancer, melanoma, ovarian cancer, classic Hodgkin lymphoma, breast cancer, colorectal cancer and prostate cancer [9,74,75] and present endogenously-derived peptides via autophagy and cross presentation [66,76]. The role of HLA class I downregulation in evading cancer-immune recognition is well established [77], yet impairment of HLA class II expression results in evasion of immune recognition independently of HLA class I [9,11]. In this study, we show that MAP1LC3C directly determines HLA class II expression in cancer cells. Cells with reduced MAP1LC3C present reduced HLA class II expression and may contribute to reduced immune recognition. For example, *in vivo* studies have shown that transduction of tumor cells with ectopic HLA class II or CIITA increases CD4$^+$ T cell and CD8$^+$ T cell-mediated tumor rejection [78–81], implying that suppressing HLA class II expression is necessary for tumor progression. In addition, selective pressure imposed by CD4$^+$ T cell on tumor cells leads to elimination of HLA class II-positive tumor cells by cognate CD4$^+$ T cells and thus outgrowth of HLA class II-negative tumor cells [82].

Although constitutively expressed on APCs, inflammatory signaling, such as IFNγ, induces HLA class II in a wide range of cell types [16]. This is primarily mediated through induction of the transcriptional key regulator, CIITA. CIITA is comprised in a multiprotein complex and is required and sufficient to induce HLA class II [13]. We provide evidence that, loss of MAP1LC3C prevents CIITA expression and consequently HLA class II expression. CIITA re-expression is enough to recover HLA class II expression, indicating that all other (known and unknown) factors that control HLA class II are functionally present within MAP1LC3C deficient cells.

CIITA and HLA transcriptional promoters are subject to epigenetic changes that determine their transcriptional expression [14]. In line with previous reports, inhibition of histone deacetylase enhances HLA class II transcription independently of CIITA [63,83].

Here we showed that MAP1LC3C (indirectly) controls HLA class II expression. Previously MAP1LC3C has been described to be essential for xenophagy to clear intracellular bacteria [34,84] and viruses [85]. Simultaneously degrading exogenous peptides and MAP1LC3C-dependent presentation on HLA class II would yield an effective and rapid immune response to infections. Although our cell culture experiments are performed under sterile conditions, distinctive microbiomes are present in human tumors [86]. Remarkably, we observed that MAP1LC3C expression correlates with HLA expression and immune cell infiltration in tumors that are predominantly presented in non-sterile organs, such as lung, esophagus, stomach, bladder, colon and rectum. This may suggest that the cancer cells, or the epithelial cell that the cancer cells are derived from, are prone to antigen presentation. Interestingly, in non-sterile tumors, intratumor bacteria infiltration has been shown to influence treatment efficacy [87,88] and patient outcome [89]. Moreover, microbiome composition has been

associated with ICI response in NSCLC patients and administration of antibiotics prior to immune therapy reduces cancer immunotherapy efficacy [90,91]. Although attempted, no associations were discovered between MAP1LC3C expression and microbiome composition in patient data (data not shown).

## Conclusion

In conclusion, we show that reduced MAP1LC3C associates with cancers that display low infiltration of immune cells ("immune cold tumors") and controls HLA class II expression through the regulation of CIITA. Hence, MAP1LC3C may be a relevant biomarker to distinguish immune responsive vs non-responsive tumors.

## Supporting information

**S1 Fig. Supporting information of s Fig 1 to S Figure 5.**
(PDF)

**S2 Fig. Raw_images.**
(PDF)

**S1 Table. Supplementary tables.**
(DOCX)

**S1 Data. Minimal data set.**
(XLSX)

## Acknowledgements

The results shown here are part based upon data generated by the TCGA Research Network: https://www.cancer.gov/tcga. The authors would like to thank Kim R. Kampen for providing chemical compounds.

## Author contributions

**Conceptualization:** Lydie M.O. Barbeau, Tom G.H. Keulers, Kasper M.A. Rouschop.

**Funding acquisition:** Kasper M.A. Rouschop.

**Investigation:** Lydie M.O. Barbeau, Nicky A. Beelen, Kim G. Savelkouls, Tom G.H. Keulers.

**Supervision:** Tom G.H. Keulers, Lotte Wieten, Kasper M.A. Rouschop.

**Writing – original draft:** Lydie M.O. Barbeau.

**Writing – review & editing:** Nicky A. Beelen, Lotte Wieten, Kasper M.A. Rouschop.

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
