## [Decision Letter · Decision Letter 0]

16 Aug 2024

PONE-D-24-21571The autophagy related gene MAP1LC3C is essential for CIITA-mediated HLA class II expression in non-small cell lung cancerPLOS ONE

Dear Dr. Rouschop,

Thank you for submitting your manuscript to PLOS ONE. After careful consideration, we feel that it has merit but does not fully meet PLOS ONE’s publication criteria as it currently stands. Therefore, we invite you to submit a revised version of the manuscript that addresses the points raised during the review process. The reviewers have raised several issues that need careful attention, including concerns about statistical analysis, inconsistencies among figures, clarity, and the alignment of results with conclusions. Please address all these comments and make the necessary revisions.

We look forward to receiving your revised manuscript.

Kind regards,

Danillo G Augusto

Academic Editor

PLOS ONE

“These studies received financial support from the Dutch Cancer Society (KWF grant 12,276 and UM2015-7735 to K.R.).”

“The results shown here are part based upon data generated by the TCGA Research Network: https://www.cancer.gov/tcga. This work was financially supported by the Dutch Cancer Society (KWF 12276 to K.R.). The authors would like to thank Kim R. Kampen for providing chemical compounds.”

“These studies received financial support from the Dutch Cancer Society (KWF grant 12,276 and UM2015-7735 to K.R.).”

6. In the online submission form you indicate that your data is not available for proprietary reasons and have provided a contact point for accessing this data. Please note that your current contact point is a co-author on this manuscript. According to our Data Policy, the contact point must not be an author on the manuscript and must be an institutional contact, ideally not an individual. Please revise your data statement to a non-author institutional point of contact, such as a data access or ethics committee, and send this to us via return email. Please also include contact information for the third party organization, and please include the full citation of where the data can be found.

Reviewers' comments:

Reviewer's Responses to Questions

**Comments to the Author**

1. Is the manuscript technically sound, and do the data support the conclusions?

Reviewer #1: Yes

Reviewer #2: No

2. Has the statistical analysis been performed appropriately and rigorously?

Reviewer #1: Yes

Reviewer #2: No

3. Have the authors made all data underlying the findings in their manuscript fully available?

Reviewer #1: Yes

Reviewer #2: Yes

4. Is the manuscript presented in an intelligible fashion and written in standard English?

Reviewer #1: Yes

Reviewer #2: No

5. Review Comments to the Author

Reviewer #1: In their manuscript Barbeau et al. analyzing TCGA datasets showed reduced expression of autophagy related gene MAP1LC3C in different type of cancers correlating with reduced HLA class II expression, infiltration with immune cells and some checkpoints on APC and T-cells. Such correlation between expression levels of MAP1LC3C, HLA class II, HLA-DMA and CIITA were confirmed by single cell transcriptomic analysis of NSCLC tumors and MAP1LC3C deficient squamous cell carcinoma. The authors demonstrated that silencing of MAP1LC3C inhabits HLA class II expression through preventing CIITA expression. The manuscript is well written, the data presented are interesting and might have clinical implications for establishing a novel biomarker for responsiveness to immunotherapy. I have some minor remarks

1) The authors should check the numbers and citation of tables in the text. For example, though the manuscript the authors refer to table 1 -page 14, line 233, 241., instead of table 6. Additionally on page 14, line 245 tables S2 and S3 are cited instead of tables S1 and S2 respectively.

2) I would recommend indicate p-values for statistical significance of Spearman correlation in table 6, tables S1 and S2 Also on figure 2G, p value should be given.

3) On page 14, line 231 authors state that there is a strong correlation between MAP1LC3C expression and HLA-DOB. However, that statement is not supported by the r value (0,3791) indicating weak correlations

4) Page 14, line 244-247 – “Comparable observation was made for LUAD and HNSCC patients”. However, although according to the data presented on Figure S2 MAP1LC3C has lower expression in both tumors, Spearman correlation in Table S1 for example show no strong correlation of MAP1LC3C expression any HLA class I, class II or check points on APC and T-cells in contrast with data for LUSC. That should be clarified in the text.

Reviewer #2: In this manuscript, the authors show some correlations between the expression of the autophagy related gene MAP1LC3C and the expression of HLA class II genes and immune infiltrations in various type of tumors. The authors then demonstrates that MAP1LC3C can modulates HLA class II and their master transcriptional regulator, CIITA at the mRNA levels. However, many of the authors claims are unsubstantiated. Based off the data presented, the authors fail to prove that MAP1LC3C is essential for CIITA-mediated HLA class II expression and fail to show that MAP1LC3C directly control HLAs and CIITA.

Major points:

The abstract talk about identifying markers for response to ICI, and as a conclusion based on their data, the authors suggest MAP1LC3C may serve as a biomarker to stratify immune responsive NSCLC. However, no data on response to ICI are presented in the manuscript. Introduction second paragraph, the statement in the first phrase is not correct. Depsite many studies showing correlations between HLA expression and ICI response, HLAs has not been shown to be required for response to ICI. For example, “MHCII-deficient tumors can have very good response to anti-CTLA-4 treatment”

https://doi.org/10.4049/jimmunol.1900778 and PMID: 30568030 . Moreover, other studies have shown that tumor volume is not affected by expression of CIITA following anti-PD-1 https://doi.org/10.1007/s00262-018-2262-5. MHCII may contribute to ICI response, but would benefit from additional investigation.

The manuscript is overall very confusing and needs to be rewritten. Some statements are not consistent with the data presented. For example, Figures 4B-E disprove the manuscript title i.e. the authors show that class II HLAs mRNAs are induced by CIITA in MAP1LC3C deficient cells. The first paragraph on page 16 is extremely confusing, and from there, we can’t understand where the authors are leading us. Better rationales are needed.

In figure 2a-G, the authors use wrong statistical analysis. The samples distribution is abnormal, Spearman correlation should be use. Also, the authors use a very low sample size to perform a correlation. Pearson or Spearman correlations necessitate at least 30 samples to provide meaningful results. This should be address.

Figure 3H is very difficult to interpret but yet very important in proving the author’s point.

First, you need to confirm that MAP1L3C3 protein level is downregulated by shRNA.

You also need a control to show that CQ treatment and MAP1LC3B knock down are efficiently blocking autophagic flux, for example by showing accumulation of p62 or ubiquitinated protein. Was CQ treatment done concomitantly with IFNg? How long after MAP1LC3B knock down? Actually, the shRNA#1 shows accumulation of HLA class II when autophagic flux is blocked by CQ.

Results in figure 4A-E are inconsistent with results in figure 3A-F. None of the genes are downregulated by MAP1LC3C shRNAs. It seems that MAP1L3C3 shRNAs did’nt work. Authors need to confirm that MAP1LC3C mRNA and protein levels are downregulated by the shRNAs.

Some references do not match with what stated in the text and many are not original research article but literature reviews, which need to be properly citated.

For example, ref 55. Also, trim28, and CHAF1A are not working with PRC2. Trim28 can interact with EZH2 in an PRC2 independent fashion and CHAF1A has been linked to transcriptional activation.

IFNg was shown to impairs autophagic flux in lung cancer cells https://doi.org/10.1080/2162402X.2021.1962591. How do they explain this? They need to show status of all protein with and without IFNg.

Parts of the discussion are completely irrelevant to the research presented in the manuscript.

Minor points:

There is no figure 2I.

Line 264 is wrong, the figure 3H does not show any data when MAP1L3C3 is depleted.

Should use bar plot in 3I and J, difficult to see.

Large number of information is missing in figure legends: target of the shRNAs, time points, drug concentrations, cell lines used.

In the material and method section, details about transfection and transductions needed. Also, give more details about generation of the MAP1LC3C deficient model and CIITA overexpression. In general, the authors should provide enough information to validate the study.

Figure 1a: add median line.

Line 282: fig.4A-F not G

6. PLOS authors have the option to publish the peer review history of their article (what does this mean? ). If published, this will include your full peer review and any attached files.

**Do you want your identity to be public for this peer review?** For information about this choice, including consent withdrawal, please see our Privacy Policy .

Reviewer #1: **Yes: ** Milena Ivanova Shivarova

Reviewer #2: No

---

## [Author Response · Author response to Decision Letter 1]

9 Oct 2024

Please also see the uploaded file as it contains also immunoblots.

We wish to thank the reviewers for their detailed assessment of our manuscript. We have carefully addressed all the comments raised and made changes to the manuscript. Please find below a detailed response to all the issues that were raised.

Reviewer #1:

In their manuscript Barbeau et al. analyzing TCGA datasets showed reduced expression of autophagy related gene MAP1LC3C in different type of cancers correlating with reduced HLA class II expression, infiltration with immune cells and some checkpoints on APC and T-cells. Such correlation between expression levels of MAP1LC3C, HLA class II, HLA-DMA and CIITA were confirmed by single cell transcriptomic analysis of NSCLC tumors and MAP1LC3C deficient squamous cell carcinoma. The authors demonstrated that silencing of MAP1LC3C inhabits HLA class II expression through preventing CIITA expression. The manuscript is well written, the data presented are interesting and might have clinical implications for establishing a novel biomarker for responsiveness to immunotherapy. I have some minor remarks

Author: We wish to thank the reviewer for the positive remarks.

1) The authors should check the numbers and citation of tables in the text. For example, though the manuscript the authors refer to table 1 -page 14, line 233, 241., instead of table 6. Additionally on page 14, line 245 tables S2 and S3 are cited instead of tables S1 and S2 respectively.

Author: We thank the reviewer for pointing this out. The correct referrals are now included in the document. We also checked all the other referrals for correctness.

2) I would recommend indicate p-values for statistical significance of Spearman correlation in table 6, tables S1 and S2 Also on figure 2G, p value should be given.

Author: In larger datasets, very small p-values can occur due to the sheer volume of data. In such cases, it’s essential to also look at the effect size (e.g., Pearson’s r), which indicates the strength of the correlation. In our datasets each correlation has a p-value that is <0.0001. As there is a high chance of overestimation in our dataset (>500 patients), we therefore focused on r values rather than the very small p-value. For completeness, we now indicated it in the table.

3) On page 14, line 231 authors state that there is a strong correlation between MAP1LC3C expression and HLA-DOB. However, that statement is not supported by the r value (0,3791) indicating weak correlations.

Author: We agree with the reviewer and have omitted HLA-DOB from this sentence.

4) Page 14, line 244-247 – “Comparable observation was made for LUAD and HNSCC patients”. However, although according to the data presented on Figure S2 MAP1LC3C has lower expression in both tumors, Spearman correlation in Table S1 for example show no strong correlation of MAP1LC3C expression any HLA class I, class II or check points on APC and T-cells in contrast with data for LUSC. That should be clarified in the text.

Author: As suggested we altered the text into: “Although reduced MAP1LC3C expression is observed in LUAD (Fig. S1H-M, Table S1) and HNSCC patients (Fig. S1N-S, Table S2), correlations with HLA and checkpoint expression are less pronounced and suggest that additional mechanisms are at play.”

Reviewer #2: In this manuscript, the authors show some correlations between the expression of the autophagy related gene MAP1LC3C and the expression of HLA class II genes and immune infiltrations in various type of tumors. The authors then demonstrates that MAP1LC3C can modulates HLA class II and their master transcriptional regulator, CIITA at the mRNA levels. However, many of the authors claims are unsubstantiated. Based off the data presented, the authors fail to prove that MAP1LC3C is essential for CIITA-mediated HLA class II expression and fail to show that MAP1LC3C directly control HLAs and CIITA.

Major points:

Reviewer: The abstract talk about identifying markers for response to ICI, and as a conclusion based on their data, the authors suggest MAP1LC3C may serve as a biomarker to stratify immune responsive NSCLC. However, no data on response to ICI are presented in the manuscript.

Author: In the abstract no referrals to development of predictive ICI biomarkers were made. The abstract concluded that LC3C may serve as a biomarker to identify immune responsive cancers. Based on the data provided, we believe that this conclusion is valid.

Although immune-cold tumors are generally less responsive to ICI, we regret that the abstract was written in such a way that the intention and the message of the manuscript was open for misinterpretation. To reduce the possibility for misinterpretation, we have carefully rewritten the abstracts content and (instead of mentioning ICI) concluded the abstract with the suggestion that cancers are selected for low MAP1LC3C expression to evade efficient immune responses.

Reviewer: Introduction second paragraph, the statement in the first phrase is not correct. Depsite many studies showing correlations between HLA expression and ICI response, HLAs has not been shown to be required for response to ICI. For example, “MHCII-deficient tumors can have very good response to anti-CTLA-4 treatment”

https://doi.org/10.4049/jimmunol.1900778 and PMID: 30568030 . Moreover, other studies have shown that tumor volume is not affected by expression of CIITA following anti-PD-1 https://doi.org/10.1007/s00262-018-2262-5. MHCII may contribute to ICI response, but would benefit from additional investigation.

Author: The reviewer is correct that HLA expression, mutational burden and antigen expression correlates with ICI efficacy, but is not essential. We have therefore altered this sentence into: “Effective immune recognition is correlated with neoantigen presentation on human leucocyte antigen (HLA) molecules”

Reviewer: The manuscript is overall very confusing and needs to be rewritten. Some statements are not consistent with the data presented. For example, Figures 4B-E disprove the manuscript title i.e. the authors show that class II HLAs mRNAs are induced by CIITA in MAP1LC3C deficient cells. The first paragraph on page 16 is extremely confusing, and from there, we can’t understand where the authors are leading us. Better rationales are needed.

Author: We have carefully edited the manuscript to reduce any confusion it may have caused and to provide improvements on the rationale for the experiments that were performed.

The data in figure 4B-E is obtained after ectopic CIITA expression, and, in our opinion, does not disprove the original title. Nevertheless, to prevent misinterpretation or confusion, we have altered the title into:

“MAP1LC3C repression reduces CIITA- and HLA class II expression in non-small cell lung cancer”

Reviewer: In figure 2a-G, the authors use wrong statistical analysis. The samples distribution is abnormal, Spearman correlation should be use. Also, the authors use a very low sample size to perform a correlation. Pearson or Spearman correlations necessitate at least 30 samples to provide meaningful results. This should be address.

Author: We agree with the reviewer that the sample size is too small for distribution assessment and that Spearman might be preferred. In addition, although we don’t think that these correlations are meaningless, the small sample size may limit statistical assessment. In the current version, for correct use and interpretation the single graphs have now been excluded. In place we have represented the data as a heat map. These data confirm the observation in the TCGA data in an independent cohort. That it is a small cohort, is explicitly mentioned in the text. As suggested, we have now also excluded correlation and p values.

Reviewer: Figure 3H is very difficult to interpret but yet very important in proving the author’s point.

First, you need to confirm that MAP1L3C3 protein level is downregulated by shRNA.

You also need a control to show that CQ treatment and MAP1LC3B knock down are efficiently blocking autophagic flux, for example by showing accumulation of p62 or ubiquitinated protein. Was CQ treatment done concomitantly with IFNg? How long after MAP1LC3B knock down? Actually, the shRNA#1 shows accumulation of HLA class II when autophagic flux is blocked by CQ.

Author: We included MAP1LC3C mRNA analyses of control and knockdown cell lines to indicate efficient knockdown (fig 3A). In line with biological processes, the large reduction in MAP1LC3C mRNA also results in an efficient knockdown of MAP1LC3C protein. As requested, we have added the immunoblots of the stable knockdown cell lines to the manuscript (Fig 3H).

In the submitted version , we included an immunoblot that indicates the efficiency of LC3B knockdown (suppl fig 2J), this is very high (>95%). As cells deficient in MAP1LC3B are defective in formation of autophagic vesicles [1, 2], in general, such an efficient knockdown would be sufficient to abrogate or, at least, inhibit autophagy. Accumulation of LC3B-II after addition of a lysosome inhibitor, such as chloroquine is a well-accepted method to determine autophagic flux [3] and has frequently be used by us in previous work e.g [4-8]. As seen previously, we confirmed that addition of chloroquine is sufficient to induce LC3B-II accumulation and proves that the concentration that we use is sufficient to block autophagic flux (Fig S2J).

In line with our previous experiences with chloroquine, we observed accumulation of p62 after CQ addition indicative of blocked flux. Furthermore, LC3B knockdown resulted in higher levels of p62 that did not further accumulate after CQ addition and confirms blocked flux (image below).

We altered the text into: “Prevention of lysosomal degradation with a sufficient dose chloroquine (CQ)) to block autophagy flux, as illustrated by LC3B-II accumulation (Fig. S2J), or genetically targeting autophagy (MAP1LC3B) [3] indicates that decreased total HLA class II protein levels are not a consequence of increased lysosomal degradation (Fig. 3H and S2J).” (p18. Line 298)

Exposure to CQ to block flux was done concomitantly with IFN exposure. As prolonged CQ exposure can induce autophagy, we also tested shorter exposures (8h and 16h). Also after shorter exposures, no differences in HLA-II processing/degradation/turnover were observed. Due to the negative nature of our results and consistency between timepoints, we decided to include only one timepoint.

To clarify the exposure times, we included: “ Unless indicated otherwise and when indicated, cells were exposed for 24 hours to IFNγ, CQ, Tazemetostat or Vorinostat. Doxycycline exposure was done for 2 days before experimentation to achieve sufficient knockdown.”

Although at first glance shRNA1 may appear to induce HLA-II flux after IFN exposure, loading differences may have leveled this effect. We believe that sufficient biological repeats are required to draw robust conclusions. As such this experiment was performed with 4, independent, biological repeats. HLA-expression was quantified using imageJ and is corrected for loading (B-actin) for each respective experiment. These quantifications indicate that HLA-II protein is reduced after MAP1LC3C knockdown and no differences are observed after CQ addition.

Reviewer: Results in figure 4A-E are inconsistent with results in figure 3A-F. None of the genes are downregulated by MAP1LC3C shRNAs. It seems that MAP1L3C3 shRNAs did’nt work. Authors need to confirm that MAP1LC3C mRNA and protein levels are downregulated by the shRNAs.

Author: The engineered cell lines with shRNA expression to target MAP1LC3C for the knockdown of LC3C express these shRNA from a, non-inducible, backbone. The plasmid to express the shRNA was lentivarally delivered and therefore integrated into the genome of the host cell. This ensures continuous and stable expression of the shRNA. Inter-experimental differences are therefore minimal. Nevertheless, all cell lines used were routinely checked for efficient knockdown, at least once a week, during execution of experiments. We have included the mRNA expression of LC3C as fig 4A.

Nevertheless, we do not agree with the conclusion of the reviewer that the data presented in Fig 4B-F is not in line with figure 3. There may be, at least, two reasons for the misinterpretation. 1) CIITA is a very potent inducer of HLA class II expression, as such the range of the y-axis increases tremendously in C, D, E and F. As such, the 2-fold differences observed in in fig 3 are “flattened”. 2) CIITA expression is under a doxycycline-inducible promoter. It is well established that these promotors may be slightly “leaky” and thus produce small quantities of CIITA, even in absence of doxycycline. This is also illustrated in fig 4B where, particularly in shRNA #2, expression of CIITA is elevated in the absence of doxycycline.

Despite these technical difficulties, what is most important are the conclusions that are drawn from these experiments. These results show that, intrinsically, MAP1LC3C-deficient cells are capable of producing HLA-II. The expression differences observed are thus not related to interference with CIITA-function, but regulated upstream at the point of CIITA –expression. In our opinion this conclusion is well supported by the provided data.

For clarification, we altered the texts into: “We therefore tested whether MAP1LC3C affects HLA class II expression through inhibition of the HLA class II key transcription factor, CIITA. For this purpose cell lines with doxycycline inducible CIITA overexpression were engineered. Although there is a minimal leakiness in the absence, addition of doxycycline elevated CIITA expression in the absence of MAP1LC3C (Fig 4 A and B). Importantly, ectopic CIITA expression rescued HLA class II and HLA chaperone (HLA-DMA) mRNA, and protein expression in MAP1LC3C deficient cells (Fig. 4 F-G). These data indicate, that although MAP1LC3C deficiency results in decreased HLA class II expression, these cells are intrinsically capable CIITA-induced gene regulation. Furthermore, these data suggests that MAP1LC3C regulates CIITA expression, either at transcription, mRNA translation or protein stability levels.” (p19 lines 316-onwards).

Reviewer: Some references do not match with what stated in the text and many are not original research article but literature reviews, which need to be properly citated.

For example, ref 55. Also, trim28, and CHAF1A are not working with PRC2. Trim28 can interact with EZH2 in an PRC2 independent fashion and CHAF1A has been linked to transcriptional activation.

Author: We agree that the factors that were identified, are not PRC2 modifiers. For correctness, we have altered the text into: “Deacetylation of histones by HDACs tightens histones interaction with DNA, resulting in gene transcription impairment [9]. HLA class II transcription is dependent on recruitment of histone modifier enzymes, such as histone deacetylases (HDACs). Analysis of the proximal interactome network [10, 11] revealed that CHAF1A, TRIM28 and PCNA interact with MAP1LC3C, recruit HDACs [12, 13] and modify HDAC activity[14].” For referencing, original research articles were included.

In the manuscript, multiple references to review articles are provided, mostly in the discussion. These refer to bodies of literature, overall perspectives, general concepts and multiple studies without discussion original ideas, methods, data, theories or direct findings. As such referencing to review articles is, to our knowledge, acceptable and were kept.

Reviewer: IFNg was shown to impairs autophagic flux in lung cancer cells https://doi.org/10.1080/2162402X.2021.1962591. How do they explain this? They need to show status of all protein with and without IFNg.

Author: It is indeed remarkable that we still observe autophagic flux after IFN-gamma addition, where Fang et al. describe inhibition of autophagic flux after IFN-gamma exposure. Our experiments were performed 4 times with independent biological repeats (Fig S2J) with each time comparable results. A potential explanation for this discrepancy, may be the use of different cell lines and cell line specific effects. This is

---

## [Decision Letter · Decision Letter 1]

17 Dec 2024

MAP1LC3C repression reduces CIITA- and HLA class II expression in non-small cell lung cancer

PONE-D-24-21571R1

Dear Dr. Rouschop,

We’re pleased to inform you that your manuscript has been judged scientifically suitable for publication and will be formally accepted for publication once it meets all outstanding technical requirements.

Kind regards,

Danillo G Augusto

Academic Editor

PLOS ONE

Additional Editor Comments (optional):

Reviewers' comments:

Reviewer's Responses to Questions

**Comments to the Author**

1. If the authors have adequately addressed your comments raised in a previous round of review and you feel that this manuscript is now acceptable for publication, you may indicate that here to bypass the “Comments to the Author” section, enter your conflict of interest statement in the “Confidential to Editor” section, and submit your "Accept" recommendation.

Reviewer #1: All comments have been addressed

2. Is the manuscript technically sound, and do the data support the conclusions?

Reviewer #1: Yes

3. Has the statistical analysis been performed appropriately and rigorously?

Reviewer #1: Yes

4. Have the authors made all data underlying the findings in their manuscript fully available?

Reviewer #1: Yes

5. Is the manuscript presented in an intelligible fashion and written in standard English?

Reviewer #1: Yes

6. Review Comments to the Author

Reviewer #1: The authors have addressed all remarks and that improved significantly the quality of the manuscript. I have no further comments

7. PLOS authors have the option to publish the peer review history of their article (what does this mean? ). If published, this will include your full peer review and any attached files.

**Do you want your identity to be public for this peer review?** For information about this choice, including consent withdrawal, please see our Privacy Policy .

Reviewer #1: **Yes: ** Milena Ivanova

---

## [Editor Report · Acceptance letter]

PONE-D-24-21571R1

PLOS ONE

Dear Dr. Rouschop,

I'm pleased to inform you that your manuscript has been deemed suitable for publication in PLOS ONE. Congratulations! Your manuscript is now being handed over to our production team.

Kind regards,

on behalf of

Dr. Danillo G Augusto

Academic Editor

PLOS ONE